# Tough Polymer Gel Electrolytes for Aluminum Secondary Batteries Based on Urea: AlCl_3_, Prepared by a New Solvent-Free and Scalable Procedure

**DOI:** 10.3390/polym12061336

**Published:** 2020-06-12

**Authors:** Álvaro Miguel, Nuria García, Víctor Gregorio, Ana López-Cudero, Pilar Tiemblo

**Affiliations:** Instituto de Ciencia y Tecnología de Polímeros (ICTP-CSIC), Calle Juan de la Cierva 3, 28006 Madrid, Spain; amiguel@ictp.csic.es (Á.M.); ngarcia@ictp.csic.es (N.G.); v.gregorio@ictp.csic.es (V.G.); ana.cudero@ictp.csic.es (A.L.-C.)

**Keywords:** polymer gel electrolytes, deep eutectic solvent, aluminum secondary batteries, self-healing, solvent-free procedure, thermoplastic

## Abstract

Polymer gel electrolytes have been prepared with polyethylene oxide (PEO) and the deep eutectic mixture of AlCl_3_: urea (uralumina), a liquid electrolyte which has proved to be an excellent medium for the electrodeposition of aluminum. The polymer gel electrolytes are prepared by mixing PEO in the liquid electrolyte at T > 65 °C, which is the melting point of PEO. This procedure takes a few minutes and requires no subsequent evaporation steps, being a solvent-free, and hence more sustainable procedure as compared to solvent-mediated ones. The absence of auxiliary solvents and evaporation steps makes their preparation highly reproducible and easy to scale up. PEO of increasing molecular weight (Mw = 1 × 10^5^, 9 × 10^5^, 50 × 10^5^ and 80 × 10^5^ g mol^−1^), including an ultra-high molecular weight (UHMW) polymer, has been used. Because of the strong interactions between the UHMW PEO and uralumina, self-standing gels can be produced with as little as 2.5 wt% PEO. These self-standing polymer gels maintain the ability to electrodeposit and strip aluminum, and are seen to retain a significant fraction of the current provided by the liquid electrolyte. Their gels’ rheology and electrochemistry are stable for months, if kept under inert atmosphere, and their sensitivity to humidity is significantly lower than that of liquid uralumina, improving their stability in the event of accidental exposure to air, and hence, their safety. These polymer gels are tough and thermoplastic, which enable their processing and molding into different shapes, and their recyclability and reprocessability. Their thermoplasticity also allows the preparation of concentrated batches (masterbatch) for a posteriori dilution or additive addition. They are elastomeric (rubbery) and very sticky, which make them very robust, easy to manipulate and self-healing.

## 1. Introduction

Energy storage systems like batteries have become a strategic market in the current wireless and hyper-connected society, and they are also a pillar of the decarbonization process. Battery demand is calculated to double every five years. As an example, the energy demand of Li-ion batteries, the most mature technology, has risen from 22 GWh in 2010 to a total expected demand of 390 GWh in 2030 [1]. The massive production of Li batteries will presumably end up by producing a shortage of the raw materials required to produce these batteries. One of the alternatives to overcome such a scenario is the development of batteries employing more accessible raw materials.

Aluminum secondary batteries can display a higher volumetric capacity than lithium (for instance in metal-air batteries ~2000 mAh cm^−3^ of lithium, and ~8000 mAh cm^−3^ of aluminum) and the raw material has better distribution and abundance in the earth’s crust (less than 0.01% vs. 8% abundance in the earth’s crust for Li and Al, respectively) [2]. Aluminum-based secondary batteries are currently one of the most appealing alternatives for energy storage. This battery requires the use of non-aqueous electrolytes to carry out aluminum stripping and plating at the anode. However, contrary to the case of the Li-ion battery, very few non-aqueous liquid electrolytes exist where the stripping and plating of aluminum (Al) have been demonstrated [3,4,5]. They are namely, the mixture of imidazolium chloride salts (EMImCl) and aluminum chloride (AlCl_3_), the deep eutectic solvents (DES) urea/AlCl_3_ and acetamide/AlCl_3_, and Et_3_NHCl/AlCl_3_ [6]. In these DES electrolytes, Al species capable of being electrodeposited are formed. Each of them has its own advantages and disadvantages, for instance urea/AlCl_3_ is non-toxic, but acetamide/AlCl_3_ is more conductive. Both DES have the advantage of being cheaper and easier to produce than EMImCl/AlCl_3_, and seem a very interesting choice because of their wide potential window and high ionic strength for the redox chemistry of metals [6,7].

Turning conventional liquid electrolytes into solid ones is a common endeavor in all metal secondary batteries, mainly because they are far safer but also because they permit more flexible geometries, and lighter devices. Battery safety is much enhanced when using solid electrolytes because they avoid leaks of toxic or corrosive liquids, and they mitigate or even eliminate the dendritic growth of the reduced metal at the anode and the subsequent short-circuits. In the case of Li-ion, which is the most mature among all-solid battery technologies, and a source of inspiration for the newest ones, porous polymer separators, dense polymer gel electrolytes and inorganic electrolytes are the most explored approaches [8,9,10,11]. Each of these has its own advantages and drawbacks, but, probably the most balanced approach is the use of polymer gel electrolytes (PGE), because of the combination of high ionic conductivity, dendrite growth mitigation, leak elimination, and mechanical endurance.

To the best of our knowledge there is only one procedure reported to date able to produce polymer gel electrolytes suitable for aluminum secondary batteries [9,10,11,12]. In this procedure, the complex acrylamide-AlCl_3_ is dissolved in dichloromethane, where it is polymerized in the presence of the ionic liquid electrolyte 1-ethyl-3-methylimidazolium chloride (EMImCl) and AlCl_3_ (EMImCl: AlCl_3_, 1:1.5, in molar ratio). The solution is casted and left overnight to dry at 60 °C. These authors studied electrolytes with different polymer ratios, and concluded that those with the higher fraction of liquid electrolyte (80 wt%) were promising Al electrolytes, which enabled the stripping and plating of Al. Other authors [13,14] have repeated the exact procedure [12], in one case, to prepare electrolytes containing 80 wt% of the same EMICl:AlCl_3_, and in another, the room temperature ionic liquid formed with triethylamonium chloride (Et_3_NHCl) and AlCl_3_ (Et_3_NHCl: AlCl_3_ 1:1.6 molar ratio). All these electrolytes were said to be solid, however no rheological measurements or qualitative characterizations of the solid state, for instance using the well-known inverted tube test, were done.

The in situ polymerization procedure involves the use of a solvent to cast the gel membranes. Sustainability and scalability require the elimination of auxiliary solvents as far as possible. Getting completely rid of the solvent is not a simple task, frequently involving heating steps, which make solvent casting cumbersome and lacking the reproducibility that solvent-free processes have, while making industrial scaling and mass production complicated. Furthermore, choosing a solvent for membrane casting is, in the case of aluminum electrolytes, by no means straightforward, as highly interactive strong Lewis acids are present in them, namely AlCl_3_ or Al_2_Cl_7_^−^, produced in the presence of an excess of AlCl_3_. As a matter of fact, the electrochemical activity of Al electrolytes (EMImCl/AlCl_3_ in particular) has been seen to decrease when diluted with solvents such as acetone, acetonitrile or THF, all of them bearing lone electron pairs [12].

This decrease occurs because of the strong interaction between the electron-deficient species AlCl_3_, and Al_2_Cl_7_^−^ present in Al electrolytes, and the lone pair of electrons in those solvents’ molecules. It has to be taken into account that Al_2_Cl_7_^−^ together with the cationic species [AlCl_2_urea_2_]^+^ [15] have been proposed to be directly related to the electrodeposition of Al in the urea:AlCl_3_ electrolytes.

Negative electrode reaction:$$4Al_{2}Cl_{7}^{-}+3e^{-}↔Al+7AlCl_{4}^{-}$$

Negative electrode reaction:$$2{\left[AlCl_{2}urea_{2}\right]}^{+}-3e^{-}\to Al+AlCl_{4}^{-}+4urea$$

For this reason, the solvent used in [12] is dichloromethane, which does not interfere in the electroactivity of the aluminum liquid electrolyte. The deleterious interaction of molecules bearing lone pairs of electrons with aluminum electrolytes has led the scientific community to rule out the use of many well-known polymers in the field of solid electrolytes for the preparation of gel Al electrolytes. This includes, for instance, polyethylene oxide (PEO), polymethylmethacrylate (PMMA) or polyacrylonitrile (PAN) [11,12,13,14]. This view is supported by experiments showing that PEO based electrolytes [9] are not able to sustain electroplating of aluminum [16]. Some electroplating is seen in electrolytes prepared with PVDF [9], but the current is certainly very low. Hence, apparently the use of common commercial polymers must be disregarded, which is a drawback because, given their solubility in Al liquid electrolytes and the strong interaction between these polymers and the species in the electrolyte, it is highly probable that polymer gels can be prepared without auxiliary solvents.

The larger the molecular weight of a polymer, the lower the wt% needed to produce a polymer gel network, and this has been studied thoroughly, for example, with PEO hydrogels [17], where it has been shown how with 4 × 10^6^ g mol^−1^ PEO in water solution, the hydrogel becomes a semidilute network over 0.63 wt% of PEO and a concentrated network over 7 wt%. Actually, our research group has employed this strategy, the use of UHMW PEO in low weight fraction, to prepare thermoplastic electrolytes for Li batteries [18,19,20,21] by melt compounding, and using organoclays as physical crosslinkers. Our hypothesis is that it is possible to produce gel electrolytes with aluminum ionic liquids and UHMW PEO, which will retain the electroactivity of the liquid electrolyte to a high extent, while employing solvent-free and scalable procedures. As the liquid electrolyte we have chosen the deep eutectic solvent formed by urea:AlCl_3_ (uralumina hereafter), which has been shown to be an excellent medium for the electrodeposition of aluminum [7].

## 2. Experimental

### 2.1. Materials

To prepare the electrolytes, PEO of molecular weights Mw = 1 × 10^5^, 9 × 10^5^, 50 × 10^5^ and 80 × 10^5^ g mol^−1^ from Sigma-Aldrich (MO, USA) was used. Uralumina150 (U150) and uralumina135 (U135), prepared with AlCl_3_:urea at a molar ratio 1.50:1 and 1.35:1 respectively, were received from Scionix Ltd. (London, UK). Uralumina is highly sensitive to humidity, producing HCl quickly when in the open air. All materials were used as received.

### 2.2. Preparation Procedure

Scheme 1 illustrates the preparation procedure. Uralumina was placed in a glass beaker on top of a heating plate, inside a glovebox under argon atmosphere ([O_2_] < 1 ppm, [H_2_O] < 1 ppm). PEO in powder was added stepwise and the mixture was stirred while the temperature increased up to 70 °C. As the temperature got close to 60 °C, viscosity changes revealed the melting and mixing of PEO, and the PGE was formed. If it was manually stirred with a glass rod, the viscosity increase could be easily felt. Table 1 shows the electrolytes studied in this work. The PEO wt% added varied from 0.7 to 5 wt%.

### 2.3. Characterization

*FT-IR spectroscopy* was used to study the structure of the final PGE, including the variation of the conformational structure of PEO in the PGEs. The electrolytes were sandwiched between 2 mm thick ZnSe windows inside the glovebox and their IR spectrum was recorded using a FT-IR Perkin-Elmer Spectrum-One, with 10 scans and resolution 4 cm^−1^. The FT-IR of neat uralumina was recorded in the same way.

*Rheological and mechanical behavior*. Because of the sensitivity to humidity of the Al-containing electrolytes, their rheology was studied inside a glovebox. Two simple procedures were employed to characterize them: the tube inversion test, very often used in gels [22], and the stretching test. For the tube inversion test, amounts of each PGE were introduced into a glass or vial, which was afterwards completely turned over. With the help of a video camera, the sample was observed for time periods ranging from several minutes to hours. Depending on the tube inversion test outcome, the electrolytes were divided into three different rheology groups. Group 1 electrolytes flowed as soon as the vial was reversed and were considered liquids. On the other end of the scale, Group 3 electrolytes did not flow even after days. An intermediate behavior was shown by some electrolytes, which did not flow on inverting the tube, but crept down the tube walls within some tens of minutes to a few hours. These electrolytes were classified as Group 2. In column 4 of Table 1 the PGEs prepared are classified into one of these three groups.

Group 2 and Group 3 gel electrolytes showed a clear elastomeric behavior, and in fact, on handling, Group 3 electrolytes behaved like rubber. They were very sticky materials, they stuck to themselves and to other surfaces. Thanks to their sticky properties, their elastomeric character was characterized inside the glovebox by a stretching test with the help of a glass rod, and their performance was recorded with a video camera. QR codes showing videos of most of the PGEs are presented in column 5 of Table 1.

To check the stability of the PGE rheology, in most of the PGEs presented in Table 1, both the tube inversion test and the stretching test, were repeated after several days of their preparation and, in some cases, even after several months.

*Electrochemistry.* To perform the measurements outside the glovebox, a lab-made cell filled with the PGE was placed into a glass recipient and sealed. The lid of the recipient was wired, and each electrode was connected before closing the isolating encapsulation. Then, the whole set-up was taken out of the glovebox to evaluate the electrochemical properties. These were evaluated in an Autolab PGSTAT 302 potentiostat/galvanostat. High purity aluminum foil (99.9999% Goodfellow, Huntingdon, UK) was used for both electrodes (working and counter electrodes) in a home-made electrochemical cell. All aluminum electrodes were cut (0.85 cm^2^) and afterwards cleaned in a 10 M KOH solution, washed by deionized water, and dried prior to all measurements. The thickness of the electrolytes in the cell was about 2 mm. Impedance measurements were carried out at an amplitude of 20 mV from 10^6^ to 1000 Hz. Conductivity was obtained from the equivalent circuit obtained after adjusting the Nyquist diagram. For the voltammetries, the same electrochemical cells were used with a third electrode as a pseudo-reference, of the same aluminum and treatment as the others. Cyclic voltammetry (CV) was carried out between −1.5 V to 1.5 V vs. Al/Al^3+^ at 20 mV s^−1^, for about 100 cycles. After about 20 cycles the steady state current was reached, as shown in Appendix A. In some cases, the cyclic voltammogram was recorded several months later, in order to assess the stability of the electrolytes.

## 3. Results and Discussion

Although the only method reported to date to prepare polymer gels with aluminum liquid electrolytes was in situ polymerization of acrylamide [12], it has also been reported that PEO was moderately soluble in other DES [23]. In comparison to in situ polymerization, simple dissolution of the polymer in the liquid electrolyte was far simpler, more reproducible and also scalable.

Initially, dissolution at room temperature of PEO 9 × 10^5^ g mol^−1^ was attempted, without success. As PEO is semicrystalline, and crystalline domains are harder than amorphous ones to swell, to dissolve the polymer it was required to raise the temperature up to 70 °C, i.e., slightly over the melting temperature of PEO which is at about 65 °C. On reaching that temperature, a very strong viscosity increase of the solution was observed. On cooling the blend, gelification was evident, with viscosity and stiffness of the final product strongly dependent on PEO molecular weight and PEO weight fraction in the gel. Table 1 collects all the electrolytes prepared, their formulation, nomenclature and some relevant physicochemical characterization: the ionic conductivitity (*σ*) of some of the electrolytes, their classification into one of the three rheology groups described in the experimental section, and their elastomeric behavior. The nomenclature chosen is the following: PEO*a*-*b*/U*c*, where *a* identifies the molecular weight of the polymer in g mol^−1^ divided by 10^5^, *b* stands for the polymer weight fraction in the gel and *c* identifies the type of uralumina employed to prepare the gel. For example, PEO50-1/U150 was a gel prepared with a 1 wt% of PEO Mw = 50 × 10^5^ g mol^−1^ dissolved in uralumina 150 (AlCl_3_:urea at a molar ratio 1.50:1).

With each molecular weight, electrolytes were prepared up to the maximum possible incorporation of PEO. For instance, with PEO 9 × 10^5^ g mol^−1^ it was not possible to dissolve over 5 wt%, since for higher wt% fractions, the large increase in viscosity occurring during the dissolution of the polymer hindered its complete dissolution. In the case of PEO 50 × 10^5^ g mol^−1^, it was probable that in the gel prepared with 5 wt% of PEO, the polymer was not completely dissolved. For PEO 80 × 10^5^ g mol^−1^ it was not possible to prepare the gel with 2.5 wt% because a non-negligible part of the polymer was not dissolved. Probably diminishing viscosity by raising the temperature over 70 °C would allow the dissolution of larger amounts of polymer, but this is not advisable since the uralumina may decompose.

### 3.1. Rheological Behavior

Not all electrolytes in Table 1 were gels, and among the gels, very different mechanical behavior could be found. The instability of aluminum electrolytes in the open air and in contact with several materials including stainless steel made the characterization of their rheological behavior a real challenge. The videos included in Table 1 illustrated well their diverse rheology by simple tests performed inside the glovebox. Making use of the inverted tube test, the PGEs were divided into three distinct rheology behaviors, as explained in the experimental section. Table 1 collects this qualitative classification. Group 1 electrolytes, which behaved like liquids, comprised PEO1-5/U150, PEO9-1/U150 and PEO50-0.7/U150. On the opposite side are Group 3 gel electrolytes, comprising PEO9-5/U150, PEO50-5/U150 and PEO50-5/U135, which behaved like rubber. All the rest of PGEs in Table 1 are classified as Group 2 electrolytes, i.e., soft elastic gels which did not flow immediately after turning the vial down but which crept down the walls after several minutes, as shown in Figure 1a for PEO50-2.5/U150. Figure 1b shows the dimensional stability of the same electrolyte PEO50-2.5/U150 as it was transferred from a beaker to a flask with the aid of a glass rod. All the gels were found to be very tough, impossible to tear, and they only broke if thoroughly stretched.

The qualitative classification of the PGEs into three rheology groups showed that the lower the molecular weight of the polymer, the higher the weight fraction required to gel the uralumina. For instance, PEO1-5/U150 was a viscous liquid (Group 1), while with the same polymer wt%, PEO9-5/U150 was a rubber (Group 3). As mentioned in the introduction, the effect of the polymer molecular weight on the shear modulus of gels prepared with the same polymer wt% fraction was well-known and was a consequence of polymer chain entanglement. The results in Table 1 indicate, on the one hand, that the gelling of uralumina with PEO followed the typical polymer gel behavior, implying that PEO was truly soluble in uralumina, with polymer coils extending in the liquid phase and entanglements between chains being produced and, on the other hand, that the interaction of the ethylene oxide units (EO) with uralumina was strong. Scheme 2 illustrates these characteristics.

Group 2 and Group 3 electrolytes behaved as elastomers, displaying large deformations on strain. Figure 1c shows different images extracted from the videos of PEO9-5/U150, which evidence the high deformation attained by these electrolytes. Videos (QR links) showing the elastomeric character of PEO9-5/U150, PEO50-1/U150 and PEO80-1/U150 appear in Table 1.

To test the thermoplastic character of these electrolytes, a 50:50 wt% mixture of solid PEO50-5/U150 and neat U150 was heated slowly up to 70 °C, stirring with a glass rod; gradually PEO5/U150 electrolyte softened, its viscosity decreased, and it was possible to mix it with the liquid U150. After 10 min at 70 °C the new diluted electrolyte (now with formulation PEO50-2.5/U150) was allowed to cool down, viscosity increased, and it was checked that no phase separation appeared. The softening of PEO50-5/U150 and subsequent dilution with U150 suggested that physical and not chemical bonding was produced in these PGEs and it evidenced its thermoplastic character. The same procedure was successfully followed with PEO50-5/U135, diluted with U135.

The fact that a concentrated gel can be diluted in such a simple way is very interesting, as it implies that it is possible to prepare PGEs from a common concentrated batch (masterbatch), a practice that increases reproducibility of samples, and a key feature of paramount importance for its potential use in industrial applications, for it allows a posteriori additivation.

From the viewpoint of materials design, the rheology of these PGEs showed very interesting results. First, the effective gelling of U150 was possible with as low as a 1 wt% of PEO, provided the polymer molecular weight was high enough. Second, the gels produced were thermoplastic, and therefore they were easy to process, mold, reshape, and recycle and could be prepared in the form of a masterbatch. Finally, when these gels were broken (which was not easy since they were very tough and elastic), they quickly stuck back together, which illustrates another key feature of these gels: they seem to be self-healing. This can be observed in the videos provided in Table 1, for instance that of PEO50-1/U150 and PEO9-5/U150, where the elastomeric and sticky character is clearly seen, or PEO80-1/U150 (seconds 17–19 especially).

Given the nature of the chemical species in uralumina and their strong avidity to interact with the polyether, it was considered a priority to check the stability (chemical, and consequently mechanical and electrochemical) of the PGEs for periods of time up to several months. The mechanical and rheological properties of these PGEs rely on the polymer chain being extremely long. Thus, reactions producing chain cleavage would lead to very conspicuous losses of viscosity and the elastic modulus of the PGEs, since the dependence of viscosity and elasticity on polymer chain length is very strong. Several PGEs were kept in the glovebox for periods ranging from several weeks to three months and their rheology features were periodically checked. During the first week after the preparation of the PGEs, an increase in viscosity and/or elastic modulus was qualitatively detected by the inversion tube test and the stretching test. After, and for at least two months, no noticeable modification of the rheological and mechanical characteristics was seen. This result is very important from a practical point of view, since these PGEs would be useless as electrolytes if their properties were lost in the short run. The rheological stability over the course of months also provided very interesting information on the chemical stability of these PGEs, i.e., the strong interaction between the ethers in the polymer chain and the uralumina species did not produce significant chain scission.

### 3.2. Structure of the Electrolytes by FT-IR

The fact that these electrolytes behaved as elastomers implied that there was a strong interaction between uralumina and PEO, which blocked the slippage of chain entanglements upon stretching of the polymer chains [24]. This strong interaction was studied by FT-IR. Figure 2 shows the FT-IR spectra of PGEs prepared with U150 and PEO of three different Mw (1 × 10^5^, 9 × 10^5^ and 50 × 10^5^ g mol^−1^), and with increasing wt%, from 1 to 5 wt%, together with the FT-IR spectrum of PEO at room temperature and that of U150. The region of 1400 to 800 cm^−1^ in PEO spectra is very sensitive to conformational changes. Interestingly, in the PGEs, the most intense FT-IR bands of solid PEO were not seen, namely the characteristic band centered at 1095 cm^−1^ (marked with a black arrow), assigned to combinations of stretching C-C and C-O, and COC deformation modes in conformation trans-gauche-trans, of the structural unit O-C-C-O [25]. There were other PEO bands which did not appear in the PGE spectra like the CH_2_ rocking at 840 cm^−1^ or the CH_2_ wagging at 1342 and 1360 cm^−1^. However, the CH_2_ twisting at 1241 and 1278 cm^−1^ was clearly seen in the PGE spectra, although slightly shifted to a higher wavenumber. Interestingly, the spectra of the PGEs showed a series of new bands in the region 1100−900 cm^−1^ (marked with red arrows in Figure 2) which were not present in either uralumina or in solid PEO.

The set of new bands in the region 1100−900 cm^−1^ was clearly proportional to the wt% of PEO, which indicated that these new bands belonged to the polymer. Figure 2 also shows that the spectra of PGEs prepared with different molecular weight PEO, but with the same PEO wt%, were very similar. Figure 3 shows the FT-IR of PEO50-5/U150 and PEO50-5/U135, together with those of PEO, U150 and U135. The FT-IR spectra showed no significant differences between the PGE prepared with U150 and U135 and 5 wt% of PEO.

The appearance of new PEO bands, together with the absence of characteristic vibrations of the PEO spectrum in the bulk or in the melt, namely those at 1150 and 1100 cm^−1^, strongly suggested that the polymer in the PGEs presented a conformational structure very different from that of bulk PEO, as could be expected. As mentioned before, the effect of the PEO molecular weight on the gel rheology indicated that the PEO coils extended in the liquid phase, and were entangled in one another, while the absence of backbone vibrations in the FT-IR suggested a very strong interaction of the chain with its surroundings. Scheme 2 depicts these gels, where segments of the PEO chain strongly interacting with the uralumina species were represented as rods, to illustrate the blocking of the backbone. Unraveling the nature of the polymer/electrolyte interactions is by no means straightforward since the speciation of uralumina is still not fully understood. Detailed investigation on the molecular structure of these gels with the help of computational chemistry tools is underway.

### 3.3. Ionic Conductivity and Electrochemistry of the Electrolytes

The purpose of this work is to evaluate whether UHMW PEO can be used to prepare aluminum gel electrolytes even if this polymer has been discarded in the literature because of its chemical interaction with acidic aluminum species. In order to evaluate the electrochemical performance of the different electrolytes and decide whether this approach to the preparation of PGEs for Al secondary batteries was feasible, cyclic voltammetry was carried out. While doing these experiments, it was soon evident that the sensitivity of the PGEs to humidity was significantly lower than that of the liquid uralumina. Nevertheless, neither of the gels could be manipulated in the open air since, in contact with ambient humidity, HCl was still produced, although at a much lower rate than in liquid uralumina. This was a positive consequence of the viscosity increase of the gels, and it made the PGEs much easier and safer to handle than the liquid electrolyte itself.

The voltammograms at the 50th cycle of some of the electrolytes in Table 1 are shown in Figure 4. The CV in U150 (orange line) showed the peaks corresponding to the oxidation at circa 1.5 V (vs. Al/Al^3+^) and electrodeposition of Al (at −1.5 V). The effect of the introduction into liquid U150 of increasing amounts, namely 1 wt%, 2.5 wt% and 5 wt%, of PEO 50 x 10^5^ g mol^−1^ is shown in Figure 4a. It can be observed that the progressive increase in the concentration of PEO in the PGE led to a decrease in the current density *j*, as compared to that of liquid U150. These effects could be easily ascribed to the combination of viscosity increase and decrease in the Al_2_Cl_7_^−^ concentration which presumably occurred on adding PEO to U150 [12]. However, it could also be seen that the only processes observed corresponded to the stripping and deposition of Al, and that the introduction of small amounts of PEO (e.g., PEO50-1/U150 or PEO50-2.5/U150) allowed the maintenance of enough electrochemical activity to consider them as potential electrolytes.

In Figure 4b the influence of the molecular weight of the polymer is presented for a 5 wt% of PEO. Please note that the current of these two samples was multiplied five-fold in order to make easier its comparison with U150. It can be observed that the decrease in electrochemical activity was more affected by the concentration of PEO than by its chain length, since only minor differences could be found in the current density for PEO1-5/U150 and PEO50-5/U150, even though the former was a liquid (rheology Group 1 in Table 1) and the latter a self-standing gel (rheology Group 3).

The fact that the rheology was far more affected than the current intensity by PEO chain length proved the success of using UHMW PEO as a strategy for the preparation of PGEs for Al secondary batteries, given that electroactive materials can be prepared with diverse rheological properties, ranging from liquidlike to solidlike by the simple procedure presented in this work. Moreover, the electrochemical properties presented in this work were stable for at least a few months, as occurred with their rheology.

Finally, in Figure 5 the variation of ionic conductivity, *σ*, and current, *j*, at a given potential (−0.8 V) is presented for the PGEs as a function of the wt% of PEO in the gels. It can be observed that the decrease in *σ* is linearly dependent on the wt% of PEO in the PGEs (dotted line), and irrespective of the molecular weight of the PEO employed, i.e., there was no influence of the PEO chain length on the final *σ* of the resulting gel. However, if instead of a gel, the blend of PEO and U150 resulted in a liquid, then *σ* was higher, as illustrated by PEO1-5/U150 (the solid black symbol).

The decrease in current density was much stronger than that of *σ*: while the latter was reduced by 6 from the most to the least conducting electrolyte in Figure 5, the former was reduced by 16. Interestingly, the decrease in the current density as a function of the wt% of PEO did not depend on the chain length of the polymer nor on whether the resulting blend of PEO and U150 was a liquid or a gel, i.e., it seemed to be only affected by the concentration of oxyethylenic units. This meant that, together with the decrease in *σ* produced by the gelling of uralumina and the subsequent unavoidable increase in viscosity, there was an additional phenomenon which decreased the electroactivity of the PGEs, most probably the chemical interaction of the ethers in PEO with electroactive Al species, as mentioned in the introduction [12].

Even if a decrease in electroactivity occurs when using PEO as a gelling polymer, Figure 5 demonstrates that these gel electrolytes can be at the same time as a solid and yet retain substantially the electroactivity of the liquid electrolyte, in this case uralumina. This has been postulated as not possible [11,12,13,14] based on the interaction between electroactive Al_2_Cl_7_^−^ species and the ether in the oxyethylenic units. Although the interaction of the oxyethylenic units in PEO reduces electroactivity of uralumina, the low amount of UHMW polymer required compensates with its effect on the speciation of the liquid electrolyte. Because of the likely nature of the interactions between the oxythylenic units and uralumina species, it can be anticipated that other aluminum liquid electrolytes will probably produce gels of a similar nature with UHMW PEO, this being a strategy with an ample field of application, and a very promising future.

## 4. Conclusions

The results shown in this work demonstrate that the use of UHMW PEO is a successful strategy to produce PGEs with uralumina as a liquid electrolyte that maintains the ability to reversibly electrodeposit and strip aluminum. These gel electrolytes are prepared by simple dissolution of the polymer in the liquid electrolyte, like conventional PEO hydrogels. This preparative strategy is simple and quick (only a few minutes), it is solvent-free and reproducible. The gels are thermoplastic, allowing the preparation of masterbatches which can be subsequently diluted and/or where additives can be incorporated making them excellent for industrial scaling and mass production. They are easy to shape and reshape, and also to recycle.

Because of the strong interactions between the UHMW PEO and uralumina, dimensionally stable gels are produced with as little as a 2.5 wt% PEO. These gels are stable for months, and no phase separation or rheological modifications are seen, if they are kept under controlled atmosphere. Their electrochemistry performance is also stable, and when CVs are repeated a few months later, the results are completely comparable. Compared to the few examples in the literature, they are able to retain significantly more ionic conductivity with respect to the liquid electrolyte with which they are prepared. Self-standing gels also retain a significant fraction of the liquid electrolyte’s current.

The sensitivity to humidity of these PGEs is significantly lower than that of liquid uralumina, improving their resistance to degradation upon accidental exposure to the open air and, hence, improving their safety. Finally, their sticky character confers them with inherent self-healing properties, which makes them very robust and easy to manipulate.

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
