# Peer review of "Tough Polymer Gel Electrolytes for Aluminum Secondary Batteries Based on Urea: AlCl3, Prepared by a New Solvent-Free and Scalable Procedure"

_polymers, 2020, doi:10.3390/polym12061336_

Round 1
Reviewer 1 Report
The current manuscript entitled, ‘Self-healing and tough polymer gel electrolytes for aluminium secondary batteries based on urea:AlCl3, prepared by a new solvent-free and scalable procedure’ systematically investigated the possibility of using new polymer electrolytes for battery applications. The manuscript is well ordered, but I have some minor suggestions and queries regarding the present work. after the issues are resolved, I would recommend this manuscript to be accepted.
- Some typing error exists in the manuscript. Please try to avoid the mistakes.
- It is better to make a schematic illustration of the preparations and structural interactions of the polymer gel electrolyte, it would better and easy understanding of the readers.
- In figure 4, the authors operated the Cyclic voltammogram of the sample PEO50-5/U150 with increasing wt% of PEO50/U150 and introduced different molecular weight PEO. But comparing graph (a) and (b), in both cases the sample PEO50-5/U150 behavior is slightly different. Why? In both cases the composition is the same, So it should be shown similar Cyclic voltammograms. Please explain this behavior.
- What about the XRD diffraction analysis of the PGE? In the results and discussion part, the author mentioned about the nature of PEO, which is semicrystalline. What happened the nature of PGE during the composition changing with different PEO and Uralumina? XRD is a good technique to study the structure, composition, and physical properties of the materials.
- There is no cycle performance using polymer gel electrolyte. It is better to compare cycling stability using different ratios of polymer gel electrolyte.
- In the reference part, reference numbers 21 and 25, please include volume and page number.
- What about the electrochemical impedance spectra polymer gel electrolytes? even though it shows good ionic conductivity.
Author Response
The current manuscript entitled, ‘Self-healing and tough polymer gel electrolytes for aluminium secondary batteries based on urea:AlCl3, prepared by a new solvent-free and scalable procedure’ systematically investigated the possibility of using new polymer electrolytes for battery applications. The manuscript is well ordered, but I have some minor suggestions and queries regarding the present work. after the issues are resolved, I would recommend this manuscript to be accepted.
- Some typing error exists in the manuscript. Please try to avoid the mistakes. The whole text has been revised.
- It is better to make a schematic illustration of the preparations and structural interactions of the polymer gel electrolyte, it would better and easy understanding of the readers. A scheme 1 of the preparation procedure has been included in the Experimental Section. No explicit reference has been done in this work to what the actual interactions between the polymer chain and the uralumina species are. Interactions between uralumina and PEO are by no means obvious, since the speciation of uralumina is still a matter of debate, and different cations, anions and even neutral species are proposed to exist. This topic is being currently studied by us in collaboration with computational chemists. We have prepared a scheme 2 showing the structure of the gel which has been introduced in the Results Section, after the FTIR study. Also, a short paragraph explaining the scheme has been added.
- In figure 4, the authors operated the Cyclic voltammogram of the sample PEO50-5/U150 with increasing wt% of PEO50/U150 and introduced different molecular weight PEO. But comparing graph (a) and (b), in both cases the sample PEO50-5/U150 behavior is slightly different. Why? In both cases the composition is the same, So it should be shown similar Cyclic voltammograms. Please explain this behavior. Dear reviewer, a representation error had occurred, which has been solved now. The sample PEO50/U150 is the same in both graphs.
- What about the XRD diffraction analysis of the PGE? In the results and discussion part, the author mentioned about the nature of PEO, which is semicrystalline. What happened the nature of PGE during the composition changing with different PEO and Uralumina? XRD is a good technique to study the structure, composition, and physical properties of the materials. These electrolytes are very sensitive to humidity and produce HCl when exposed to open atmosphere. They also corrode stainless steel, this is why characterization is very complex and routine techniques cannot be used. However, if we could do XRD, I doubt we would find any PEO crystals in these PGE since the results strongly suggest that the polymer is well dissolved. In particular, the absence of bulk conformations in FTIR seem to point to the good dissolution of the polymer.
- There is no cycle performance using polymer gel electrolyte. It is better to compare cycling stability using different ratios of polymer gel electrolyte. This work deals with the use of UHMW PEO as a means to circumvent the well-known drawback of this polymer in its use in the preparation gel Al electrolytes. This drawback (deleterious interaction with aluminium species) affects not only PEO but also polymers like PAN, PVDF, PEO or PMMA, all of them employed for the preparation of polymer based electrolytes for Li batteries. According to other authors (reference 12 in the manuscript) these commercial polymers cannot be used in the preparation of aluminium gel electrolytes because of their basic character. These polymers are extremely convenient as gel formers because they are commercial and cheap, and if they can be used to prepare gel aluminium electrolytes by simple mixing as proposed in this work, it will largely simplify the Al gel electrolyte panorama. This work’s main aim is to show that if UHMW PEO is employed, it is possible to lower its concentration to a point where the Al plating occurs to a similar extent as in the liquid electrolyte. We believe this is successfully shown in the CV curves presented. We are aware that the performance on cycling in a two-electrode cell is of most importance for its application in batteries. It is ongoing work and the full characterization in a two-electrode set up with dedicated cathodes is being carried out in collaboration with other partners. However, our point here is to prove that there was electrodeposition of Al in the gel electrolytes prepared by the proposed way and to show what the influence of PEO molecular weight is.
- In the reference part, reference numbers 21 and 25, please include volume and page number. Both these references have an article nº, but no page nº.
- What about the electrochemical impedance spectra polymer gel electrolytes? even though it shows good ionic conductivity. EIS was used merely to obtain the value of ionic conductivity, though what is really interesting here are the CV curves of the gels. As explained in the manuscript, other authors (reference 12) have shown that ether containing molecules interact with acidic species Al2Cl7- in aluminium electrolytes, producing AlCl4- instead. Such a change in the species concentration may not be reflected in conductivity (which is proportional to the viscosity increase), but would be seen in the CV curves. The CV curves in figure 4, in particular those prepared with the lower concentration of PEO, resemble those of liquid uralumina and prove that the ability of the gels to strip/plate aluminium is maintained if the concentration of PEO is kept sufficiently low, what can be achieved if UHMW PEO is used. All gels tested achieve the steady state after about 20 cycles and keep their properties for at least 100 cycles. Since the CV curves are the most relevant results of this work, together with rheology, after reading the reviewer’s comments we have considered it interesting to show the evolution with cycles of both uralumina and the most viscous gel PEO50-5/U150 between 1-100 cycles. This has been included as supplementary information, if considered pertinent.
Reviewer 2 Report
The authors reported a gel polymer electrolyte for Al-batteries based on UHMW polyethylene oxide and the DES uralumina. The gel polymer electrolyte was prepared by a solvent free method and characterized by FT-IR spectroscopy, cyclic voltammetry and impedance spectroscopy. Rheological properties are investigated using the inverted tube test. My recommendation is to accept this work in the present form.
Author Response
Some modifications have been done following the comments of other reviewers
Reviewer 3 Report
Dear Authors,
The manuscript titled `Self-healing and tough polymer gel electrolytes for aluminium secondary batteries based on urea:AlCl3, prepared by a new sol-vent-free and upscalable procedure`is a well presented manuscript. The results are encouraging and the approach is applicable. However, there are some concerns to be addressed:
- The manuscript needs a thorough reading to avoid the wrong sentence connections and some grammatical errors.
- Authors have presented "Depending on the tube inversion test outcome, the electrolytes have been divided into three different rheology groups. Group 1 electrolytes flow as soon as the vial is reversed and are considered liquids. On the opposite, Group 3 electrolytes will not flow even after days and are considered as solids" however, the naming system seems strange and calling a system that has no flow to a solid is very confusing, authors may use quasi solid or tough gel something like that.please avoid the term solid.
- The claim for self healability should be scientifically confirmed. A self healing material should be functioning without the human invention. in this case, it seems not the case. hence such claim should be checked again.
- The CV curves looks rather strange with many side reaction where extra current is coming, indeed the baseline looks not straight and the discussion part regarding this is not sufficient. Also it is not clear where the Al is deposited or extracted from the curve.
- Authors should demonstrate the capability of Aluminium deposition, the amount of Al deposited, number of cycles, its Coulombic efficiency etc. however, such data is missing here.
- The stretchability or other similar properties should be validated by using mechanical analyses using DMA, or rheology measurements.
- A scheme of PGE and its chemical interactions will give more information to the reader, especially, as a conclusion of FTIR analysis.Due to these concerns, my recommendation is MAJOR revision.Regards
Author Response
The manuscript titled `Self-healing and tough polymer gel electrolytes for aluminium secondary batteries based on urea:AlCl3, prepared by a new sol-vent-free and upscalable procedure`is a well presented manuscript. The results are encouraging and the approach is applicable. However, there are some concerns to be addressed:
- The manuscript needs a thorough reading to avoid the wrong sentence connections and some grammatical errors. We have gone through the text and tried to correct it.
- Authors have presented "Depending on the tube inversion test outcome, the electrolytes have been divided into three different rheology groups. Group 1 electrolytes flow as soon as the vial is reversed and are considered liquids. On the opposite, Group 3 electrolytes will not flow even after days and are considered as solids" however, the naming system seems strange and calling a system that has no flow to a solid is very confusing, authors may use quasi solid or tough gel something like that.please avoid the term solid. Throughout the manuscript, the term solid has been changed into solid-like or gel.
- The claim for self healability should be scientifically confirmed. A self healing material should be functioning without the human invention. in this case, it seems not the case. hence such claim should be checked again. These gels are self-healing since they repair themselves without human intervention. In fact, if torn or broken (something which occurs only if extremely stretched, since these gels are very tough and elastic) they will stick back. We should have provided videos showing the restoration of a damage, such as a cut, but because of COVID we are not able to produce experiments showing this in a reasonable time period, so we have preferred to eliminate the term self-healing from the title and simply suggest it as a possibility in the text, in view of the stickiness and elasticity shown in the videos.
- The CV curves looks rather strange with many side reaction where extra current is coming, indeed the baseline looks not straight and the discussion part regarding this is not sufficient. Also it is not clear where the Al is deposited or extracted from the curve. The CV in this work are the typical aluminium oxidation/reduction curves (see for example “High Coulombic efficiency aluminum-ion battery using an AlCl3-urea ionic liquid analogue electrolyte” Michael Angell, Chun-Jern Pana, Youmin Rong, Chunze Yuan, Meng-Chang Lin, Bing-Joe Hwang and Hongjie Dai,834–839 | PNAS | January 31, 2017 | vol. 114 | no. 5 ). The CVs presented in Figure 4 are the 50th of 100 cycles. The purpose of this electrochemical characterization is to show that, if PEO of sufficiently high molecular weight is used, it is possible to prepare gels which retain the electrochemical activity of the liquid electrolyte (in this case uralumina) to a high extent, and thus, that this preparation methodology is apt for the preparation of aluminium gel electrolytes. This is what figure 4 shows and what is discussed in that section. Finally, we conclude that gels with 1% or 2,5wt% of PEO 5x106 g/mol are worth testing in real batteries with dedicated cathodes.
- Authors should demonstrate the capability of Aluminium deposition, the amount of Al deposited, number of cycles, its Coulombic efficiency etc. however, such data is missing here. As mentioned in the previous point, this work deals with the use of UHMW PEO as a mean to circumvent the well-known drawback of this polymer in its use in the preparation gel Al electrolytes. This drawback (deleterious interaction with aluminium species) affects not only to PEO but also to polymers like PAN, PVDF, PEO or PMMA, all of them employed for the preparation of polymer-based electrolytes for Li batteries, and all of them useless according to other authors because of their basic character (reference 12 of the article). These polymers are extremely convenient as gel formers because they are commercial and cheap, and if they can be used for gel aluminium electrolytes, as we have done, using a simple and scalable preparation procedure, it will largely simplify the Al gel electrolyte outlook. In summary, the main aim of this work is to show that if UHMW polymer chains are employed, it is possible to lower the concentration of PEO to a point where the plating occurs to a similar extent as in the liquid electrolyte. We believe this is successfully shown. We continue to work extensively with electrolytes as those presented in this work. In collaboration with partners of the SALBAGE consortium, dedicated to Al-S batteries, we are working on the study of two-electrode cell set up with specific cathodes where of course the complete study suggested by the reviewer will be carried out.
- The stretchability or other similar properties should be validated by using mechanical analyses using DMA, or rheology measurements.Dear reviewer, we perform very complete rheological and mechanical characterizations when working with Li electrolytes. Also, we characterize not only ionic conductivity, but also diffusivity in all Li electrolytes (see for instance our references 19-21). However, aluminium electrolytes are a completely different topic. These electrolytes are very difficult to characterise since they produce HCl in contact with the open air, and they corrode stainless steel. The typical characterization routines which we all use in our laboratories like rheological curves or dielectric spectroscopy cannot be used here. We believe we have been able to produce sufficient characterization as to prove the main point in this work, i.e., the successful preparation of aluminium gel electrolytes using UHMW PEO, combining self-standing ability and retaining electrochemical activity comparable to uralumina. We are trying to adapt techniques to produce quantitative rheological measurements, since we will continue working on this topic in the future, but as for now it is out of the scope of this work.
- A scheme of PGE and its chemical interactions will give more information to the reader, especially, as a conclusion of FTIR analysis. We have prepared a scheme 2 illustrating what we know of the gel so far, which has been introduced in the Results Section, after the FTIR study. Also, a short paragraph explaining the scheme has been added. However, no explicit reference to what the actual interactions between the polymer chain and the uralumina species has been done in this work. Interactions between uralumina and PEO are by no means obvious, since the speciation of uralumina is still a matter of debate, and different cations, anions and even neutral species are proposed to exist. This topic is being currently studied in collaboration with computational chemists. We hope the scheme helps understand the morphology of the gels described in the work.
Round 2
Reviewer 1 Report
The revised manuscript is now acceptable for publication. the authors have elaborated properly to all the comments.
Reviewer 3 Report
Dear Authors,
The changes made according to the recommendation is satisfactory. Hence I would like to recommend the acceptance of the manuscript in its current form. Good luck to the Authors.
Regards,